# The Multiple Challenges of Nutritional Microbiome Research During COVID-19—A Perspective and Results of a Single-Case Study

**DOI:** 10.3390/nu16213693

**Published:** 2024-10-30

**Authors:** Anna Donkers, Waldemar Seel, Linda Klümpen, Marie-Christine Simon

**Affiliations:** Nutrition and Microbiota, Institute of Nutrition and Food Science, University of Bonn, 53115 Bonn, Germany

**Keywords:** COVID-19, hygiene, microbiota, nutritional behavior, pandemic

## Abstract

The global coronavirus disease 2019 (COVID-19) pandemic has affected multiple aspects of people’s lives, which may also influence the results of studies conducted during this period across diverse research domains. This particularly includes the field of nutritional science, investigating the gut microbiota as a potential mediator in the association between dietary intake and health-related outcomes. This article identifies the challenges currently facing this area of research, points out potential solutions, and highlights the necessity to consider a range of issues when interpreting trials conducted during this period. Some of these issues have arisen specifically because of the measures implemented to interrupt the spread of small acute respiratory syndrome coronavirus 2 (SARS-CoV-2), while others remain relevant beyond the pandemic.

## 1. Introduction

The term gut microbiota refers to the diverse community of microorganisms residing in the gastrointestinal tract, consisting of bacteria, archaea, viruses, and fungi [1,2,3]. Research over the past decades has demonstrated the importance of the intestinal microbial community in various aspects of human health [4]; thus, studying the mutualistic relationship between microorganisms and hosts [5] has become a rapidly growing field of research. In this context, the recognition that diet has a significant impact on the gut microbiome [6,7,8,9] has led to an increasing number of studies investigating the targeted modulation of gut bacteria through nutritional interventions. However, studying this interaction is a challenging task because of the complexity and dynamics pertaining to dietary intake and the gut microbial composition. Analysis of the intricate interplay between the gut microbiome and nutrition was further compounded in late 2019, when the first cases of a then-unknown upper respiratory tract infection surfaced in Wuhan, China. The disease spread rapidly, and by January 2020, the cause was identified as a novel coronavirus, SARS-CoV-2 (small acute respiratory syndrome coronavirus 2), and the disease was named COVID-19 (coronavirus disease 2019). A few months later, in March 2020, the World Health Organization (WHO) officially labeled it as a pandemic [10]. Early in the pandemic, the approaches that different countries used to deal with the situation varied widely. Many of them were policies aiming to reduce the probability of contracting and spreading the virus, such as lockdowns and social distancing [11]. The newly emerged situation had a profound effect on different areas of research. With an increased focus and need for rapid COVID-19 research in order to develop effective treatments and vaccinations, other study fields were at greater risk of being slowed down or interrupted because of limited supplies and capacities. In particular, with regard to microbiome research, it was proposed that the pandemic will have far-reaching effects on the gut microbiota resulting from lifestyle changes brought about by the restrictive measures, as well as increased hygiene practices such as frequent hand washing and the use of sanitizers [12]. While the WHO declared an end to the international health emergency for COVID-19 in May 2023, questions remain unanswered about the impact of the disease and the implemented measures, which are important for interpreting the results of studies conducted during the COVID-19 period and for future research. This perspective aims to illustrate the difficulties that were encountered during the COVID-19 pandemic in microbiome research from the viewpoint of nutritional sciences and provide suggestions for dealing with these challenges. In addition, we present the findings of a single-case study that specifically investigates the effects of hygiene practices on the gut microbiome.

## 2. Methods

### 2.1. Literature Research

To identify relevant articles for this perspective, a literature search was conducted focusing on the effects of social isolation, hygiene measures, SARS-CoV-2 infections, and COVID-19 vaccinations on gut microbiota, human metabolism, and dietary habits during the COVID-19 pandemic. The search was completed by two researchers according to standard procedures in the PubMed database. The key search terms included ‘COVID-19’, ‘SARS-CoV-2’, ‘COVID-19 vaccination’, ‘pandemic’, ‘lockdown’, ‘hygiene’, ‘sanitizer’, ‘disinfectant’, ‘diet’, ‘nutrition’, ‘gut microbiota’, and ‘gut microbiome’. Original articles and systematic reviews in English were included. Only studies on humans were selected, with a focus on adult populations and outcomes particularly relevant to nutrition research, such as metabolic disorders (e.g., obesity and type 2 diabetes) and associated clinical parameters. The initial search was conducted in November 2021 and was updated in April 2024.

### 2.2. Case Study

During our initial literature search, we identified a research gap regarding the impact of hygiene measures implemented during the COVID-19 pandemic on gut microbiota. Therefore, we conducted a single-subject case study to investigate the exclusive impact of hygiene practices on the composition of the fecal microbiome. The studied subject was a 25-year-old, healthy, normal-weight female student without former SARS-CoV-2 infection and three times vaccinated (Pfizer BNT162b2 mRNA vaccine) before the start of the intervention. The last vaccination took place three months prior. This study was conducted from May to July 2022, when all lockdown measures were lifted. The subject followed the typical hygiene practices recommended during the pandemic for four weeks without any further behavioral changes. During the four-week intervention period, commercially available alcohol-based hand sanitizer and disinfectant soap were used at least five times a day. During the two-week washout period, the hygiene measures were discontinued, and the subject returned to a regular hygiene regime. Importantly, to study the sole effects of the hygiene intervention, the participant was advised to keep dietary intake as well as exercise habits constant to maintain stable body weight. Serial fecal samples, 19 samples in total, were collected throughout the intervention and washout periods to study the gut bacterial diversity and taxonomic composition. The intervention period was divided into three phases, with the first phase including all samples collected during the first week, the second phase including samples from weeks two and three, and phase three including all samples from week four. Samples were frozen at −80 °C within 24 h after collection. Fecal DNA extraction and sequencing were performed according to a standard protocol, as described in detail elsewhere [13]. Briefly, total genomic DNA was extracted from 120 mg fecal material using ZR BashinBead lysis tubes (Zymo Research, Freiburg, Germany) and the chemagic DNA stool kit (Perkin Elmer, Rodgau, Germany) according to the manufacturer’s instructions with a mechanical lysis step added. Next-generation sequencing of the V3-V4 region of the 16S rRNA gene was performed on the Illumina MiSeq platform. Obtained data were preprocessed in QIIME 2 [14], including a quality control step using the DADA2 plugin. Sequences were then classified using SILVA databases to identify amplicon sequencing variants (ASVs) for sequences with >99% similarity. Alpha and beta diversity metrics were calculated in QIIME 2 from rarefied counts with an even sampling depth of 29,256. Data were subsequently imported into R (version 4.3.0) and combined into a phyloseq object [15]. The obtained ASVs were aggregated to the phylum and genus levels and converted into relative abundances and centered-log ratio transformed, respectively, using the microViz package for visualization [16].

## 3. Changes in Dietary and Lifestyle Habits

Most people’s lives changed dramatically with the restrictions of the lockdown. The closures of work offices, restaurants, and recreational and sports facilities increased time in the home environment and inevitably led to changes in lifestyle and dietary behavior, as already reported in several studies. Of concern, in a cohort of elderly people in England, those with obesity were more likely to engage in adverse behaviors, including decreased physical activity and increased food intake, which was associated with an increase in body weight [17]. Similarly, in a survey conducted during the first social lockdown in the United Kingdom, individuals with a highly elevated BMI, in particular, reported a perceived negative change in terms of beneficial dietary and activity behaviors. Further, a higher BMI was associated with poorer diet quality, overeating, and lower physical activity. This indicates that adults with a higher BMI may be at the greatest risk of increased weight gain during the pandemic [18]. Indeed, in a longitudinal observational study of individuals with metabolic disorders, adverse changes in dietary habits and related shifts in a substantial proportion of the cohort from overweight to obesity were observed during the first social lockdown [19]. The data indicate that individuals at high metabolic risk, who are often the focus of nutritional intervention studies, were more likely to engage in behaviors that resulted in unfavorable changes with respect to weight and body mass during the pandemic. However, perceived negative effects on health behaviors may not apply to everyone, as shown in a retrospective longitudinal observational cohort study of 896,286 participants, in which individuals with initially less desirable health behaviors improved their diet quality and weight in the course of the pandemic compared with those reporting healthier behaviors [20]. In line with these findings, a slight overall improvement in dietary quality, as quantified by the Healthy Eating Index, was demonstrated, especially in individuals with obesity [21].

A similar inconclusive pattern is seen in systematic analyses. On the one hand, less favorable nutritional behavior, including increased snacking and alcohol consumption, signs of disordered eating, and higher sedentary times, were observed as a result of the pandemic restrictions [22,23,24,25]. On the other hand, although individual behaviors such as snack frequency worsened, the overall diet quality improved slightly on a global scale [26]. Thus, the observed variability in behavioral changes may be attributed to individual responses to the pandemic conditions. While involuntary changes due to the lockdown situation affected all people in a similar manner, those resulting from conscious behavioral decisions or in response to emotions experienced because of the circumstances of the pandemic are person-specific. An analysis of a cross-sectional online survey during the first social lockdown period in Germany identified three patterns of change in eating habits and food intake among the population—a constant pattern, a health-orientated pattern, and an emotionally driven pattern. In the first pattern, only minor shifts were observed, for example, spending more time on food preparation, while the second health-orientated pattern was characterized by deliberate selection or avoidance of specific foods with attention to the consumption of healthy foods. For people with the third pattern, emotions had a large impact on eating behavior, and higher intakes of less favorable foods such as sweets and alcohol were reported. Female gender, overweight, obesity, and an immigrant background were among the factors that were associated with assignment to one of the changing patterns [27]. Similar diverging patterns were also observed in other populations. In a cross-sectional survey, participants with perceived unhealthy dietary changes attributed them to emotional states such as boredom and stress. In the participants with more favorable changes, these were mostly due to the intention to strengthen their immunity against contracting COVID-19 [28]. Alarmingly, among individuals with prediabetes, the majority reported that the pandemic negatively affected stress and anxiety levels. In addition, a decreased motivation to implement or maintain healthy habits was evident, especially in females [29].

Given the observed inter-individual variability in lifestyle and dietary changes, it is difficult to draw definitive conclusions regarding the direction and extent to which the pandemic has influenced behavior. The large heterogeneity between the studies’ methodologies and the lack of validated methods specifically geared towards an exceptional situation such as the pandemic further exacerbated the challenge of synthesizing the evidence. Especially in the context of microbiome research, the observed alterations in eating behavior and lifestyle, regardless of their direction, posed a considerable challenge as they may have resulted in significant changes in the gut microbiome and other outcome variables of interest when conducting nutritional studies. For instance, a decline in diet quality during lockdown phases may have led to a decrease in gut microbiota diversity, along with a comparatively increased abundance of *Bacteroides* species, which is commonly associated with a Western-style diet characterized by excess energy intake and high consumption of refined carbohydrates and fats. Conversely, an improvement in nutritional quality, for example, by increasing the consumption of whole plant foods, may have promoted the growth of beneficial bacterial species, such as *Bifidobacterium* spp. and *Lactobacillus* spp., capable of producing short-chain fatty acids by fermenting dietary fibers [30]. Indeed, a comparison of independent samples collected before and during the pandemic revealed compositional differences in the gut microbiome, which may be a consequence of the described changes in lifestyle and also of a changed hygiene behavior [31]. This might have a profound effect when studying microbiome–host interactions in controlled trials and is particularly relevant in nutritional studies, where expected effect sizes are relatively small. The premise of participants maintaining their habitual behavior to isolate and quantify the intervention effect often did not hold during periods of lockdowns, re-openings, and rapidly changing regulations. In addition, the individual’s ability to comply with the intervention recommendations may also have been influenced by pandemic-related mental health issues [32]. Thus, close monitoring of participants’ behavior, especially under pandemic conditions, was necessary to detect any non-compliance with the study protocol. However, in nutrition research, measuring compliance is a difficult task even under non-pandemic conditions since the use of dietary assessment methods and questionnaires surveying physical activity for this purpose can introduce errors, as they are subjective methods and susceptible to self-reporting bias. The most frequently used methods include 24-h recalls and food records [33], which are also commonly employed to determine adherence to dietary instructions in intervention studies [34]. Those methods are valuable tools in nutrition sciences and can provide useful approximations of dietary intake at the group level but lack precision at the individual level [35]. The information obtained may be insufficient to measure individual compliance to an intervention as it only covers a short period of time, making it difficult to capture variation along the study period that existed as a result of the changing pandemic regulations. Repeated assessments are therefore essential to ensure that food intake is recorded as accurately as possible to improve the quality of outcome conclusions. The use of newer technologies, such as activity trackers and continuous glucose monitoring coupled with software applications in which participants record their consumed meals and sports throughout the whole intervention period, allow for more detailed recording and individual assessment. However, these methods also have drawbacks, such as the generation of large amounts of data, which increases accuracy but also requires specific analytical expertise. Furthermore, they do not necessarily overcome the bias that also arises from traditional methods, such as reactive behavior, but could even reinforce it if the body’s blood glucose response or the number of steps is directly visible to the participant. Measurement of food-specific blood or urinary biomarkers, where available, are objective parameters that complement self-reporting methods, and their combined use is valuable to verify that participants were able to adhere to the intervention protocol during lockdown periods. If the measurement of biomarkers was not originally planned, the collection of sufficient reserve samples for subsequent use for this purpose was an important advantage. To date, such biomarkers are only available for a limited number of specific food components, and biomarker profiles for dietary patterns have yet to be developed [36]. It is therefore important to sensitize researchers to the difficulties posed by the pandemic and thus further raise awareness of the importance of carefully and predictively designed studies. Eventually, in the long term, this may lead to an improvement in existing methods and the development of new technologies that allow for greater validity of the results of nutritional studies, e.g., by extending continuous measurements by sensors beyond glucose levels and identifying novel nutritional biomarkers.

## 4. Infection and Vaccination

Although COVID-19 is primarily a respiratory disease, the findings that the cell entry receptor of SARS-CoV-2 is expressed in the gastrointestinal tract [37] and viral RNA is excreted by a large proportion of infected individuals via the feces [38] suggest that the virus may also exert effects on the intestinal compartment including the microbiome. In addition, during the course of COVID-19, some individuals exhibit gastrointestinal symptoms such as diarrhea, abdominal discomfort, nausea, or vomiting [39], which can also manifest as part of post-acute sequelae of COVID-19 or long COVID [40].

The available evidence has demonstrated that SARS-CoV-2 infection induces a state of gut microbial dysbiosis, including loss of diversity and depletion of beneficial species, alongside an increase in opportunistic pathogens (Table 1) [41,42,43,44,45,46]. These compositional changes have also been shown to be associated with an impaired metabolic capacity of the COVID-19-related gut microbiota, including alterations in amino acid and carbohydrate metabolism and a reduction in the production of short-chain fatty acids, which can even persist post-recovery [47,48]. Short-chain fatty acids can modulate immune function by activating G-protein-coupled receptors and inhibiting histone deacetylases, promoting the secretion of anti-inflammatory cytokines and suppressing pro-inflammatory pathways [49]. Consistent with this, the reported COVID-19-induced disruptions in gut microbial function were associated with disease severity and the inflammatory response, including elevated levels of pro-inflammatory cytokines and CXCL-10, a chemokine characteristic of COVID-19 [47,48]. High-risk individuals for severe COVID-19 include patients with metabolic disorders like type 2 diabetes, who show notable differences in microbial and metabolite composition compared with healthy controls [50], potentially contributing to susceptibility to increased disease severity. Interestingly, SARS-CoV-2 infection led to greater changes in the microbial composition of patients with type 2 diabetes compared with people without and thus to increased gut microbiota dysbiosis [51,52]. Most studies investigating the gut microbiome in COVID-19 were conducted in hospitalized patients, including those with critical disease, raising the question of whether the gut microbiome is also altered in patients with non-severe COVID-19. Indeed, one study showed that the evenness and community structure of the gut microbiome differs from that of healthy controls, even in mild and asymptomatic COVID-19 patients, which is still observed after negative conversion of SARS-CoV-2 RNA [53]. Other studies found no differences in the overall microbial structure in mild and moderate disease; however, taxonomic changes and a destabilized microbiota were reported [54,55]. Moreover, alterations in the gut microbiome have been found in post-acute COVID-19 syndrome. For example, compared with healthy controls, the composition of individuals with long COVID remained distinct after virus clearance, which persisted also after 6 months. In participants with no signs of long COVID, the gut microbiome returned to resemble that of non-COVID controls more closely, and no differences in species were detected 6 months after recovery [56]. Another study on recovered patients one year after infection demonstrated that individuals experiencing long COVID symptoms displayed lower alpha diversity and differences in taxonomic composition compared with non-COVID-19 controls. In asymptomatic individuals, diversity and taxonomy were mostly similar to the control group, but at least some taxonomic differences remained. For example, the genus *Erysipelotrichaceae* UCG-003 and its corresponding family *Erysipelatoclostridiaceae*, as well as *Coprococcus*, were persistently reduced in both recovered groups [57].

Besides changes in the gut microbiome, SARS-CoV-2 infection can induce metabolic alterations, which also interact with COVID-19 progression [58] and, therefore, affect relevant outcome parameters of interest in nutritional studies. For example, distinct lipidomic signatures between patients with acute disease and convalescent patients indicate that during recovery, shifts in metabolic lipid profiles occur, with specific correlations between differential lipid and fecal microbiome features [59]. Another study showed that COVID-19 induced insulin resistance as well as dyslipidemia in previously metabolically healthy individuals and that these alterations persisted after viral elimination. Of note, two short-chain fatty acids, propionic and isobutyric acid, were thereby elevated in serum from patients and positively correlated with HOMA-IR, suggesting that SARS-CoV-2-associated changes in the secretion of gut microbiome metabolites may potentially be involved in the systemic metabolic alterations observed in COVID-19 states [60].

In addition to the impact of the infection itself, it is also of interest to consider whether the administration of the COVID-19 vaccine may induce shifts in the composition of the gut microbiota, particularly in the context of research conducted during the pandemic. Interestingly, Ng et al. found decreases in alpha diversity and shifts in beta diversity compared with baseline one month after the second vaccine dose, regardless of whether an inactivated or an mRNA vaccine was administered [61]. At the taxonomic level, both vaccines induced reductions in several bacterial species, although in the mRNA vaccine group, the structural changes were more pronounced as a larger number of taxa were affected. Similar observations were made when vaccinated individuals were compared with unvaccinated individuals, the first displaying lower diversity and a higher Firmicutes-to-Bacteroidetes ratio. Furthermore, a vaccine-associated bacterial signature that included enrichment in *Faecalibacterium* was reported [62]. In addition, along with an altered microbial composition, changes in microbial functions, especially an increase in amino acid metabolism, and fecal metabolomic profiles were observed in a case-control study comparing healthy adults after COVID-19 vaccination with unvaccinated controls [63]. In contrast, an observational study investigating the temporal impact of COVID-19 vaccination on the gut microbiome in healthy and immuno-compromised individuals revealed that the intestinal microbiome, including diversity, taxonomy, and function, remained stable after the vaccination independent of the initial microbiota, immunology, and vaccine response [64].

Although the impact of infections and vaccinations on the gut microbiome and metabolism is not fully understood yet, they have impacted ongoing studies and their respective results. As the pandemic progressed, studies could experience increased dropout rates due to rapidly increasing numbers of individuals with infections, and it became impossible to include only individuals who had not previously contracted COVID-19. Here, however, it was unknown how long potential effects from infection would persist, making it uncertain what constituted an appropriate time period between recovery and enrollment in a study. Furthermore, the impact of vaccination shortly before or during the study period was and is still not fully known. Study populations during the pandemic thus consisted of a mixture of recovered, vaccinated, and previously uninfected, unvaccinated individuals with a risk of uneven distribution across intervention groups. This heterogeneity in the study population could significantly affect the results of nutritional intervention studies since recovered and vaccinated individuals may have had a different starting point, making it difficult to discern whether any changes existed as a result of recovery from disease or vaccination or as a true effect of the intervention. Therefore, for the publication of studies conducted during pandemic research times, comprehensive assessment and reporting of the individuals’ prior infection and vaccination status should be provided. The stratification of participants in bioinformatic analyses can be useful in detecting potential influences, and it provides an opportunity to study how dietary interventions could support improvements in the microbiome and metabolic alterations in subjects with previous COVID-19 infection. For future studies, the new knowledge about the potential instability of the microbiome due to infections and vaccinations should be taken into account when defining exclusion criteria for enrolling participants in studies where a sufficiently stable state of the gut microbiome is desired. However, there are currently still no established reference periods between illness or vaccination and stabilization of the microbiome composition. Preliminary indications from a small case study suggest that a semi-stable state of the gut microbiome may be present either immediately or up to 4 months after virus clearance [65]. In comparison, following antibiotic administration, a period of 1.5 to 2 months is typically reported for the general structure of the microbiome to recover [66,67], though individual taxonomic changes may persist longer [68]. Hence, antibiotic use within the last 3 months is commonly defined as an exclusion criterion in microbiome studies. Although antibiotics are likely to have a greater impact on the gut ecosystem, they could serve as a first point of reference. Therefore, further study is needed for a more accurate understanding of the effects of acute respiratory infections such as COVID-19 and vaccination on the microbiome, particularly on how long potential effects persist and how they relate to clinical outcomes, in order to derive precise recommendations.

## 5. Effect of Pandemic Hygiene Measures on the Gut Microbiota

It has been proposed that exaggerated hygiene in modern societies can affect the microbial composition and diversity of the gut, contributing to a rise in the prevalence of metabolic diseases like obesity [69]. Indeed, alteration of the gut microbiome composition resulting from high exposure to household cleaning agents in early infancy has been shown to potentially mediate an increased risk of overweight later in childhood [70]. This raises the question of how the acute increases in hand hygiene during the pandemic impact the bacteria in the intestinal tract. To examine the isolated effect of hygiene measures on the gut microbiome, we conducted a single-subject case study.

During the study period, no medication was taken, and no vaccinations or acute infections occurred. The subject’s body weight remained stable throughout the observation period (phase 1: 56.0 kg, phase 2: 56.1 kg, phase 3: 56.0 kg, washout: 56.2 kg). A reduction in alpha diversity was observed in the second phase after the hygiene measures were implemented, best reflected by observed features, but also apparent in Shannon entropy and Faith’s phylogenetic diversity (PD). Thereby, decreases in Faith’s PD indicated that phylogenetic richness was reduced across the board rather than a specific narrow phylogenetic group. In the third phase, a slight increase was observed, and during washout, baseline diversity was restored (Figure 1A). Principal Coordinates Analysis based on Jaccard distances showed distinct clusters of the samples collected during the different interventional phases, indicating a shift in bacterial composition (Figure 1B). Furthermore, a change in taxonomic composition over the course of the intervention was also evident. At the phylum level, the proportion of Firmicutes decreased with prolonged time of product use, accompanied by an increase in the relative abundance of Bacteroidetes and Actinobacteria (Figure 1C). An altered Firmicutes-to-Bacteroidetes ratio was one of the first indications of a link between the microbiome and obesity [71], and although it is an overly simplistic measure to identify a potential dysbiosis, it is still often used in (nutritional) studies to characterize the gut microbiota. At the genus level, a decline in several genera became evident at the beginning of the second phase of the hygiene intervention, with genera like *Lactococcus* and *Solobacterium* making up a lower relative amount in the bacterial composition of the samples. It is well established that lactococci are capable of producing antimicrobial peptides (bacteriocins). Consequently, a reduction in these bacteria could have a negative effect on the colonization resistance of the intestinal microbiota. It has recently been demonstrated that bacteriocins produced by *Lactococcus lactis* are effective against clinical and food pathogens [72]. The only known species of the genus *Solobacterium*, *Solobacterium moorei*, is suspected of being a marker for poor oral hygiene [73]. A reduction in this bacterium may indicate that the hygiene measures employed also had a positive effect on oral hygiene, which in turn could be discernible in the gut microbiome [74]. In contrast, *Coprobacillus*, *Faecalibacterium*, and *Bifidobacterium* showed a pattern of increased abundance during the later phases of the intervention. Moreover, an increased abundance of *Streptococcus* was observed, representing a potentially pro-inflammatory bacterial marker (Figure 1D). Among the changed taxa, typical short-chain fatty acid producers like *Faecalibacterium* and *Bifidobacterium* have been repeatedly reported to be altered in diet-related diseases like type 2 diabetes [75], making them a potential target for their treatment. Therefore, understanding the response observed is of particular interest to nutritional studies. It may be the case that an effect of hygiene measures could interfere with the conclusions of microbiome intervention studies in general. Interestingly, changes in diversity and some taxonomic changes started to recover during the washout phase and resembled the conditions at the beginning of the intervention, suggesting that the effect of hygiene measures is rather quickly reversible. However, the extent to which changes in diversity and taxonomy will persist if these measures are implemented over a longer period and if this translates to changes in metabolic parameters is uncertain. Although restricted in generalizability because of the single-subject design, these results provide preliminary evidence that the systemic changes in hygiene practices during the COVID-19 pandemic could result in relevant changes in the intestinal bacterial community. Varying degrees of implementing hygiene measures by individual participants might introduce heterogeneity within an investigated study population, which consequently complicates the interpretation of results in the setting of a nutritional interventional study, where only small effect sizes are expected, and limits comparability between studies. It is therefore important to assess study results conducted during the pandemic under this knowledge and to take individual hygiene practices as a possible confounder into account in microbiome research in general.

## 6. Conclusions and Outlook

Evidence to date shows that during the COVID-19 pandemic, heterogeneity in study populations might have emerged because of changed behavior, infection, and vaccination status and differences in the magnitude of the use of hygiene products, resulting in greater complexity and reduced comparability of study findings. Prolonged changes in the gut microbiome composition and functionality were reported after SARS-CoV-2 infection and vaccination, which might interfere with other clinical and exploratory outcomes of nutrition studies. The hygiene practices that accompanied the pandemic potentially induce relevant changes in the gut microbiome, as our pilot single-case study tentatively suggests, which might be clarified in detail by larger studies. Future analyses should consider the intestinal microbiome as a functioning ecosystem, including fungi and viruses, which interact with each other and the host. This could help to understand potential interactions between the microbiome, metabolism, and respiratory diseases in order to differentiate intervention from disease or recovery effects. Thus, for studies conducted during the pandemic, potential prior infections, vaccinations, and hygiene practices should be considered in subsequent analyses of study results, for example, by means of sensitivity analyses. However, particularly relevant to dietary intervention studies with a relatively small number of participants and effect sizes, calculated sample sizes based on pre-pandemic study results may not adequately detect diet-induced differences because of additional sources of variability during the pandemic, potentially limiting power for subgroup analysis. Despite these challenges and unresolved questions, this does not render the results of studies conducted during this time invalid; however, special care may be required in their interpretation as well as the sensitization of researchers for that topic even beyond the pandemic. The scientific community invests great efforts to capture the difficulties and changes caused by the pandemic, which will set the obtained research results in the right context. As a consequence, this might be seen as an opportunity for the development of new technologies and research strategies, especially in the transdisciplinary research field investigating the interaction between nutrition and the gut microbiota.

## Figures and Tables

**Figure 1 nutrients-16-03693-f001:**
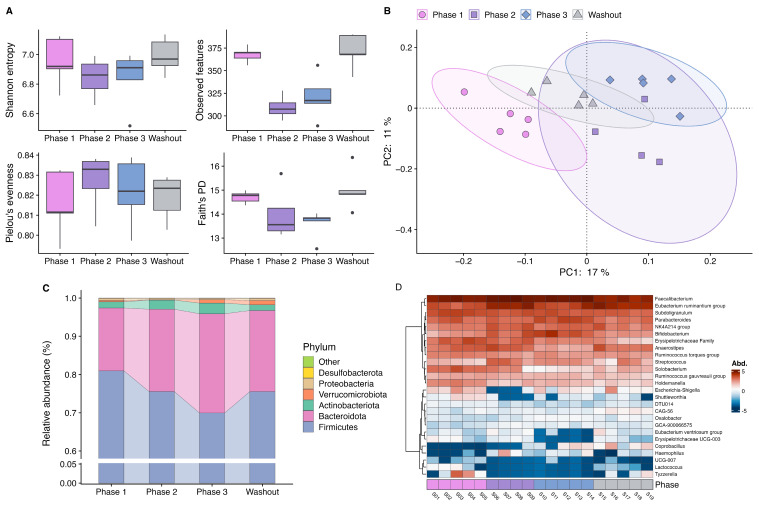
Results of the single-case study investigating the effects of hygiene measures on the gut microbiome during the 6-week intervention (phase 1: samples from week 1, phase 2: samples from weeks 2 + 3, phase 3: samples from week 4, washout: samples from weeks 5 + 6). (**A**) Boxplots of alpha diversity metrics displaying first, second (median) and third quartiles. Circles represent outlier. (**B**) Principal Coordinates Analysis based on Jaccard distance. (**C**) Barplot showing relative abundances of phyla. (**D**) Heatmap of centered log-ratio transformed abundances of bacterial taxa at the genus level. Abbreviations: PD, phylogenetic diversity.

**Table 1 nutrients-16-03693-t001:** Most commonly reported alterations in the relative abundance of genera in the fecal microbiome due to COVID-19 compared to controls according to systematic reviews. Of note, consistent increases in opportunistic pathogens and decreases in beneficial bacteria, including short-chain fatty acid producers, were observed, while results for *Bifidobacterium* were controversial.

Increase	Decrease
*Streptococcus* [42,43,44,46]	*Faecalibacterium* [42,43,44,46]
*Enterococcus* [42,43,46]	*Roseburia* [42,43,45]
*Bacteroides* [42,43,45,46]	*Coprococcus* [42,43]
*Bifidobacterium* [44]	*Bifidobacterium* [43,44,46]
	*Ruminococcus* [42,43,44,46]
	*Lachnospira* [42,43,44,45,46]
	*Prevotella* [42,43]
	*Dialister* [42,43]

## Data Availability

The raw data supporting the conclusions of this article will be made available by the authors upon request.

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
