# Peer review of "The Multiple Challenges of Nutritional Microbiome Research During COVID-19—A Perspective and Results of a Single-Case Study"

_nutrients, 2024, doi:10.3390/nu16213693_

Round 1
Reviewer 1 Report
Comments and Suggestions for Authors
This great perspective shows the effects of COVID-19 on the intestinal microbiome, as well as the studies concerning it. The text is well worked out, showing effects on everyday's life. The case report's Figure is quite clear and informative.
I would propose to add a small Table, showing the bacteria involved before and after COVID-19.
extra comment: This perspective needs a Discussion paragraph, in order to interpret the Results correctly.Author Response
Please see the attachment.

Reviewer 2 Report
Comments and Suggestions for Authors
The manuscript is very interesting. The authors provide information that may be very useful regarding the role of the microbiota in inflammation and defense capacity. However, I have the following comments.
I. Major comments:
1. Reading the manuscript makes sense. However, the structure is confusing. Initially it seems like a review and then it is presented as a results paper (section 4). I think it would be better to present it as a review.
2. I suggest including a section (methodology) after the introduction that reports the criteria used to select the cited papers.
3. What clinical projection would this work have, for example: interventions with foods that can modulate the microbiota?
4. Regarding the microbiota and the inflammatory response, how could short-chain fatty acids influence it?
5. Improve the wording of the objective of the study.
6. How does a diet high in energy, refined carbohydrates and fat affect the microbiota?
Round 2
Reviewer 2 Report
Comments and Suggestions for Authors
Authors answered all my comments. The manuscript was improved. Therefore, the manuscript can be accepted.